# Prediction of Mismatch Repair Status in Endometrial Cancer from Histological Slide Images Using Various Deep Learning-Based Algorithms

**DOI:** 10.3390/cancers16101810

**Published:** 2024-05-09

**Authors:** Mina Umemoto, Tasuku Mariya, Yuta Nambu, Mai Nagata, Toshihiro Horimai, Shintaro Sugita, Takayuki Kanaseki, Yuka Takenaka, Shota Shinkai, Motoki Matsuura, Masahiro Iwasaki, Yoshihiko Hirohashi, Tadashi Hasegawa, Toshihiko Torigoe, Yuichi Fujino, Tsuyoshi Saito

**Affiliations:** 1Department of Obstetrics and Gynecology, Sapporo Medical University of Medicine, Sapporo 060-8556, Japan; tort.39fore.antares@gmail.com (M.U.); takenakayuka.sksm@gmail.com (Y.T.); saishinsyou@gmail.com (S.S.); motoki.gyne@gmail.com (M.M.); miwasaki@sapmed.ac.jp (M.I.); tsaito@sapmed.ac.jp (T.S.); 2Department of Media Architecture, Future University Hakodate, Hakodate 041-8655, Japan; g3221001@fun.ac.jp (Y.N.); g2122046@fun.ac.jp (M.N.); fujino@fun.ac.jp (Y.F.); 3Gomes Company LLC, Sapporo 004-0875, Japan; t-horimai@gomes-company.co.jp; 4Department of Surgical Pathology, Sapporo Medical University of Medicine, Sapporo 060-8556, Japan; ssugita@sapmed.ac.jp (S.S.); hasetada@sapmed.ac.jp (T.H.); 5Department of Pathology, Sapporo Medical University of Medicine, Sapporo 060-8556, Japan; kanaseki@sapmed.ac.jp (T.K.); hirohash@sapmed.ac.jp (Y.H.); torigoe@sapmed.ac.jp (T.T.)

**Keywords:** endometrial cancer, deep learning, artificial intelligence, biomarker, mismatch repair, molecular classification, whole-slide imaging, digital pathology

## Abstract

**Simple Summary:**

In the context of endometrial cancer, molecular classification is becoming increasingly significant as the molecular class determines appropriate treatments and the prognosis. However, performing clinical testing for molecular classification in a large number of patients entails significant financial and time costs. Therefore, there is a need to develop a substantial molecular profile screening method for endometrial cancers. The objective of this study was to explore whether the molecular classification of endometrial cancer could be predicted from digital images of hematoxylin and eosin (H&E)-stained slides using deep learning as a screening tool. After making adjustments to the training data set and hyperparameters, we confirmed the feasibility of estimating the mismatch repair status from histological digital images of endometrial cancer. Deep learning was found to be effective for predicting one aspect of the molecular classification from H&E-stained histological digital images.

**Abstract:**

The application of deep learning algorithms to predict the molecular profiles of various cancers from digital images of hematoxylin and eosin (H&E)-stained slides has been reported in recent years, mainly for gastric and colon cancers. In this study, we investigated the potential use of H&E-stained endometrial cancer slide images to predict the associated mismatch repair (MMR) status. H&E-stained slide images were collected from 127 cases of the primary lesion of endometrial cancer. After digitization using a Nanozoomer virtual slide scanner (Hamamatsu Photonics), we segmented the scanned images into 5397 tiles of 512 × 512 pixels. The MMR proteins (PMS2, MSH6) were immunohistochemically stained, classified into MMR proficient/deficient, and annotated for each case and tile. We trained several neural networks, including convolutional and attention-based networks, using tiles annotated with the MMR status. Among the tested networks, ResNet50 exhibited the highest area under the receiver operating characteristic curve (AUROC) of 0.91 for predicting the MMR status. The constructed prediction algorithm may be applicable to other molecular profiles and useful for pre-screening before implementing other, more costly genetic profiling tests.

## 1. Introduction

The lifetime risk of women developing endometrial cancer is approximately 3%, and, over the past 30 years, the overall incidence has increased by 132%, reflecting an increase in risk factors (particularly obesity and aging) [1]. Endometrial cancer is classically classified into two groups—namely, Type I or II tumors [2,3]. Type I endometrial tumors are associated with excess estrogen, obesity, hormone receptor positivity, and abnormalities in hormone receptors. On the other hand, Type II tumors, which are mainly serous, are often observed in older, non-obese women and are considered to have a worse prognosis [2,4].

In recent years, there has been a growing focus on molecular biological classification of endometrial cancers. The classification of endometrial cancer proposed by The Cancer Genome Atlas (TCGA) in 2013—a joint project of the National Cancer Institute (NCI) and National Human Genome Research Institute (NHGRI)—employed milestone data for the molecular classification of endometrial cancer [5]. TCGA proposed classification in four classes—POLE (ultramutated), MSI (hypermutated), Copy-number-low (endometrioid), and Copy-number-high (serous-like)—based on next-generation sequencing (NGS) data obtained from 232 cases of endometrial cancers. Following TCGA classification, Talhouk et al. developed and verified a modified molecular classification method called “ProMisE” [6,7]. This method replaces the detection of an abnormality of the *TP53* gene and microsatellite status, which are dependent on sequencing, with immunohistochemical staining (IHC), making molecular classification of endometrial cancers more clinically accessible. The ProMisE method classifies endometrial cancer into four molecular sub-types: POLEmut, Mismatch Repair Deficient (MMRd), p53abn, and NSMP (No Specific Molecular Profile, p53wt). These four classes correspond to the POLE, MSI, Copy-number-low, and Copy-number-high classes of TCGA, respectively. In 2020, the World Health Organization (WHO) also recommended a molecular classification for endometrial cancer [8,9]. In 2023, the molecular-biology-based classification of the Federation of Obstetrics and Gynecology (FIGO) staging was also demonstrated [10]. Therefore, it is anticipated that the provision of a stable and easy method for molecular profiling of endometrial cancer will become clinically significant in the near future.

Based on these molecular classification results, one of the most crucial therapeutic agents to consider for classification-matched treatment is the immune checkpoint inhibitor (ICI). ICIs are being investigated and gaining interest for various type of tumors, including endometrial cancer [11]. Programmed death receptor-1 (PD-1) is an immune checkpoint molecule expressed on activated T-cells, with programmed cell death ligand 1 (PD-L1) being a representative ligand [12]. PD-1 and PD-L1 inhibitors accelerate cancer cell elimination, mainly mediated through cytotoxic T-cells. One of the accepted surrogate markers for the effectiveness of ICIs is deficient mismatch repair (dMMR) and the resulting microsatellite instability (MSI) [13]. According to a cross-organ analysis of solid tumors, endometrial cancer had the highest frequency of MSI, occurring at a frequency of 17% [14].

Commonly used methods for determining dMMR/MSI status are based on polymerase chain reaction (PCR) [15,16], IHC for MMR proteins [17], and NGS [18,19]. The use of IHC for MMR status classification involves examining the expression of MutL homolog 1 (MLH1), MutS homolog 2 (MSH2), MutS homolog 6 (MSH6), and post-meiotic segregation increase 2 (PMS2) [17]. In endometrial cancer, the high detection rate of dMMR underscores the utmost importance of immunologic profiles [14,20]; however, testing for detailed molecular profiles (including MMR status) in every endometrial cancer patient can be expensive and time-consuming, which could complicate the course of treatment. Therefore, it is necessary to develop alternative molecular classification methods in order to reduce the associated financial and time costs.

As a novel classification method, we focused on approaches using machine learning. In the field of healthcare, deep learning has already been demonstrated to be useful for the classification of medical images. Originally, deep learning emerged as a prominent sub-field of machine learning [21]. There have been many reports on the effective use of deep learning approaches for image classification in clinical use, such as magnetic resonance imaging (MRI) of the brain [22], retinal images [23], and computed tomography (CT) of the lungs [24], among others [25]. Unlike conventional machine learning methods, deep learning relies on deep neural networks, which mimic the operation of the neurons in the human brain. Deep learning networks can automatically extract the significant features necessary for the corresponding learning tasks with minimal human effort [21].

Among the various deep learning algorithms, convolutional neural networks (CNNs) have been the most commonly used [26]. Each convolutional layer extracts different information and, through stacking multiple convolutional layers, the network can progressively extract more complex and abstract features. The activation function in the middle of the convolutional layers enhances the network’s ability to handle non-linear problems and adapt to different distributions [27]. The network may gain features that humans may not be consciously aware of; however, many of these features can be challenging to articulate. Some reports [28,29,30] in the field of cancer have suggested that hematoxylin and eosin (H&E) staining can be used to predict genetic alterations and features of the tumor microenvironment without the need for further laboratory testing. In particular, determining microsatellite instability (MSI) status by deep learning analysis of H&E-stained slides has been described in several reports focused on gastric cancer [31] and colon cancer [32,33,34,35,36]. In endometrial cancer, Hong et al. [28] attempted a comprehensive assessment using deep learning for the detection of histological subtypes and genetic alterations, achieving an area under the receiver operating characteristics curve (AUROC) ranging from 0.613 to 0.969 despite variations in the assessment criteria. Additionally, Fremond et al. [30] similarly attempted to carry out decision-making through the use of deep learning approaches and to visualize the histological features specific to molecular classifications in endometrial cancer.

Thus, the use of artificial intelligence—particularly deep learning—for medical image analysis has been rapidly expanding [25,37]. Therefore, we considered the potential application of deep learning to address issues related to endometrial cancer. In this study, we examined the utility of CNNs and novel attention-based networks for the prediction of the MMR status of endometrial cancer.

## 2. Materials and Methods

### 2.1. Ethical Compliance

According to the guidelines of the Declaration of Helsinki, informed consent was acquired through an opt-out form on the website of Sapporo Medical University. The Sapporo Medical University Hospital’s Institutional Review Board granted approval for this study under permission number 332–158.

### 2.2. Patients and Specimens

For this study, formalin-fixed paraffin-embedded (FFPE) tumor samples from Sapporo Medical University Hospital were used. Surgical specimens were obtained from patients with a primary site of endometrial cancer. A pathologist and a gynecologic oncologist chose representative slides of endometrial cancer resection specimens that were stained with H&E. Of the 127 patients with endometrial cancer treated from 2005 to 2009 in total, we excluded 7 patients with non-endometrioid cancer and 6 patients without sufficiently available tumor component tiles (Figure 1). Of the 7 non-endometrioid cancer cases, 4 were serous cancer cases and 3 were clear cell cancer cases. These 7 non-endometrioid cases were excluded from this study due to the following issues. Firstly, the number of specimens was insufficient to construct individualized classification models for each cancer. Secondly, these non-endometrioid cancers significantly differ morphologically from endometrioid cancer and may have resulted in bias in the classification models if included as part of the overall data set.

### 2.3. Immunohistochemistry Staining and Evaluation of MMR Status

FFPE tumor tissues were cut into 4 μm slices, and Target Retrieval Solution at pH 9 (DAKO, Glostrup, Denmark) was used for epitope retrieval. The tissues were then stained with rabbit anti-MSH6 monoclonal antibody (clone, EP51; DAKO) and mouse anti-PMS2 monoclonal antibody (clone, ES05; DAKO), which were used to detect MMR proteins in the tissues. The slides then underwent incubation with a secondary antibody. Subsequently, the slides underwent hematoxylin counterstaining, followed by rinsing, alcohol dehydration, and cover-slipping with mounting medium. Two gynecologists and one pathologist evaluated the resulting IHC and MMR status. As previously reported, negative staining for MSH6 corresponds to a lack of MSH2 and/or MSH6 proteins, as the stability of MSH6 depends on MSH2 [38]. In the same way, PMS2 staining covers the protein expression of PMS2 and/or MLH1. Therefore, if either PMS2 or MSH6 expression was deficient, it was determined as dMMR and, if not, it was determined as proficient MMR (pMMR) (Figure 2B). In total, 29 patients were classified as dMMR, while 85 patients were classified as pMMR (Figure 1).

### 2.4. Pre-Processing of Whole-Slide Images

The H&E slides were then digitized using a Nanozoomer whole-slide scanner (Hamamatsu Photonics, Hamamatsu, Japan). Each whole-slide image (WSI) was divided into non-overlapping square tiles of 942 μm at a magnification of 5×, each with dimensions of 512 × 512 pixels (Figure 2A). On average, each WSI was divided into 813 tiles, and processing WSIs from 120 cases of endometrioid cancer resulted in the creation of 97,547 tiles.

We first constructed an image exclusion program, in which we specifically conducted the following pre-processing steps using OpenCV: (i) excluding edge tiles with different numbers of pixels in height and width and (ii) converting the tile to HSV format and binarizing the tile through treating pixels that matched the specified pink color range from (100, 50, 50) to (179, 255, 255) as white (255) and pixels that did not match as black (0). The program calculated the average value of the pink color area and excluded it if it was greater than 25 (i.e., if there was a large amount of pink color within one tile). In total, 38,699 tiles were excluded automatically using the constructed program. Furthermore, we manually excluded 53,451 tiles without sufficient tumor component. The exclusion criteria were specified as follows: tiles in which more than 25% of the tile area consists of non-tumor components (e.g., stroma), tiles containing irrelevant contaminants within the slide, tiles with folding due to poor tissue extension during sample preparation and air trapping, and tiles with artifacts during scanning. Appendix A shows an overview of the tile exclusion process through the program and manual inspection. The total number of excluded tiles (Appendix A) amounted to 92,150, while eligible tiles (Appendix A) amounted to 5397, accounting for 5.5% of the total number of divided tiles. Appendix A details the number of tiles and characteristics for each patient.

### 2.5. Hardware and Software Libraries Used

The experiments were carried out with Python (version 3.8.10), making use of the following packages: torch (version 2.0.0), torchvision (version 0.15.1), numpy (version 1.24.1), scikit-learn (version 1.2.2), matplotlib (version 3.7.1), and timm (version 0.6.13). Model development and evaluation were performed on a workstation with GeForce RTX 3080 (NVIDIA, Santa Clara, CA, USA) graphic processing units, a Ryzen Threadripper 3960X (24 cores, 3.8 GHz) central processing unit (Advanced Micro Devices, Santa Clara, CA, USA), and 256 GB of memory.

### 2.6. Data Split and Training Data Preparation

The useful tiles were divided into separate data sets for training, validation, and testing. The data set cases were randomly split into training, validation, and test sets for each prediction task, such that tiles from the same patient were contained in only one of these sets. This approach ensured that the test data set was independent from the training process, allowing for a patient-level split. The split ratio for training–validation–testing was set at 70%:15%:15%.

### 2.7. Classification Model Construction Using Convolutional Neural Networks

Construction of the CNN-based binary classification model for MMR status, pMMR, or dMMR was conducted using pre-trained CNN models through torchvision in the Pytorch library, including GoogLeNet [39], VGG19 [40], ResNet50 [41], ResNet101 [41], wideResNet101-2 [42], and EfficientNet-B7 [43]. We constructed a model that inputs a non-overlapping image tile of size 512 × 512 pixels at a resolution of 1.84 μm/pixel and outputs a tile-level probability for MMR status. We fine-tuned the pre-trained models in torchvision using the prepared training data set and validated the results using the validation data set, following the provided instructions. The trainable parameters were fine-tuned using a stochastic gradient descent optimization method, and we examined the conditions for data pre-processing and the hyperparameters needed for model training. To address the imbalance in the number of tiles in each class, we down-sampled the larger class of pMMR, randomly reducing cases to align with the smaller class in terms of slide numbers. The detailed results of the down-sampling process are presented in Appendix A. We also examined changes in model performance with data augmentation. We conducted the following four patterns of data augmentation: (i) no data expansion (original tile), (ii) original tile with added 90° and 270° rotations (resulting in three times the data), (iii) original tile with added vertical and horizontal flips (resulting in four times the data), and (iv) original tile with both rotations (as in ii) and flips (as in iii) (resulting in six times the data). Furthermore, we examined the conditions for the hyperparameters, provisionally using ResNet50 [41] for the validation network. For the hyperparameters, we changed the batch size (8, 16, 32), number of epochs (30, 60, 90, 120), and learning rate (1 × 10^−2^, 1 × 10^−3^, 1 × 10^−4^).

### 2.8. Classification Model Construction Using Attention Networks and Our API-Net-Based Model

We also verified the performance differences between CNNs and attention-based networks, such as a Vision Transformer (ViT) [44]. We selected pre-trained ViT models from the torchvision models in the Pytorch library, as mentioned above. The hyperparameters and data set were similarly chosen as for the CNNs mentioned above. We examined two ViT models—ViT_b16 and ViT_b32—in this study. Additionally, we examined the model of the modified network based on API-Net [45]. This modified network is a class-aware visualization and classification technique that employs attention mechanisms, which we developed for cytopathological classification and feature extraction. This API-Net-based model takes pairs of images as input and learns the embeddings of input features and representative embeddings, called prototypes, for each MMR class. We used the existing API-Net to estimate attention vectors. Given an unknown image, the classification model predicts classes through comparing the unknown images to prototypes, recognizing their similarity for the determination of classes.

### 2.9. Evaluation of Constructed Model Performance

The following calculated parameters were used as indicators of model performance: Accuracy = (TP + TN)/(TP + FP + FN + TN); Precision = TP/(TP + FP); Recall = TP/(TP + FN); and F-score = 2 × precision × recall/(precision + recall). TP, TN, FP, and FN represent the number of true positive, true negative, false positive, and false negative tiles, respectively. A receiver operating characteristic (ROC) curve is a probability curve for classification of problems at various threshold settings. The ROC curve was plotted using TPR against FPR, where TPR is on the *y*-axis and FPR is on the *x*-axis. The AUROC represents the area under the ROC curve.

## 3. Results

### 3.1. Pre-Processing of Data Set before Model Training

Table 1 shows the results of the data set pre-processing before model training. First, we examined the ratio of the number of tiles between data sets. Appendix A shows the number of tiles regarding different data set ratios. When the ratio of the number of tiles was not adjusted for predicting the MMR status, the pMMR class had approximately 2.6 times the amount of data as the dMMR class. Specifically, there were 1484 tiles for dMMR and 3913 tiles for pMMR.

Next, we created a data set using down-sampling in order to match the number of slides for pMMR to the lower slide count of dMMR and compared it with the original ratio without adjustment. In the down-sampled data set, the number of tiles for dMMR remained unchanged (at 1484), while for pMMR, it was 1484. Consequently, the original ratio seemed to be better when examining accuracy values exclusively. As a result, in the original ratio, the classification results were biased toward pMMR, which had a larger number of tiles. In other words, due to the increase in FN and TN, the recall rate was 0.09 and the precision rate was 0.55, both of which are low. Compared to the original data set, the down-sampled data set exhibited superior overall performance. Therefore, it was revealed that the classification performance improved in the data set with down-sampling, even though much of the pMMR training data were excluded. Additionally, we examined the effect of training on a data set that had undergone data augmentation through flipping and rotating processes, but no improvement in performance was observed. We used ResNet50 for the consideration of these effects on data set pre-processing.

### 3.2. Validation of Model Performance in Various Hyperparameter and Classification Models

Table 1 shows the hyperparameter tuning results. To predict the MMR status, we used a down-sampled data set with good performance and adjusted the hyperparameters. Compared to batch sizes of 16 or 32, a batch size of 8 showed good accuracy results; therefore, we selected a batch size of 8. Regarding the number of epochs, we adopted 30 epochs as, in our study, the model prediction performance in the validation data set presented high values up to epoch 30, regardless of whether a greater number of epochs was considered. Regarding the learning rate, we chose 1 × 10^−2^, which performed better than even lower values. We conducted a hyperparameter search for the API-Net-based model and adopted a learning rate of 1 × 10^−3^, which yielded the highest accuracy.

Table 2 shows the results regarding the differences among the classification models. We examined the differences among the classification models without changing hyperparameters between CNNs and ViTs. We utilized pre-trained networks available in Pytorch (version 2.0.0) and focused on validating the classical networks commonly used for CNNs. In all CNNs, the AUROC exceeded 0.8, with particularly high values (0.89) observed for ResNet50 and ResNet101. On the other hand, in the comparison and evaluation of attention methods, satisfactory performance was not achieved with any ViT model (AUROC: ViT_B16, 0.62; ViT_B32, 0.76). However, regarding our developed API-based network, the model achieved sufficient performance, with 0.81 accuracy and 0.89 AUROC.

### 3.3. Performance of the Models for Unseen Test Data Set

Next, we examined the generalization performance of the model through adapting it to untested data sets. Figure 3 shows the results regarding the performance of the best-performing model on the test data set, while Appendix A shows the performance of all models validated on the test data set. We investigated the best methods for the pre-processing of the data set, hyperparameters, and classification models, as described above. As a result, we adopted a combination of down-sampling in the ratio between the two classes. For CNNs, with the combination of batch size = 8, epochs = 30, and learning rate = 1 × 10^−2^, utilizing ResNet50 or ResNet101 yielded the best performance. Among the attention methods, our API-Net-based model achieved the best accuracy when compared to that of the other pre-trained ViT models.

## 4. Discussion

The incidence of endometrial cancer is increasing worldwide [46] and, considering the rising importance of molecular biological tests, we need to think about future approaches to diagnosis and treatment in this context. The molecular profile of endometrial cancer generates a large number of features that can be utilized to provide information about treatment. The TCGA-based molecular classification and its modified classification algorithm of ProMisE are representative prognostic biomarkers and helpful for choosing classification matched therapies, as mentioned above [5,6]. Additionally, the PORTEC-RAINBO [47] trial is one of the largest clinical trials investigating genotype-matched therapy for endometrial cancer, which aims to improve clinical outcomes and reduce the toxicity of unnecessary treatments in patients with endometrial cancer through molecularly directed adjuvant therapy strategies. One of the RAINBO trials, the MMRd-GREEN trial, enrolled patients with dMMR endometrial cancer at stage II with significant lymphovascular space invasion (LVSI) or stage III, mismatch repair deficient endometrial cancer. It then compared a group receiving adjuvant radiotherapy with concurrent and adjuvant durvalumab for one year with a group receiving radiotherapy alone. Assessment of MMR status will become increasingly important in the future, and in this trial, IHC was used for the determination of dMMR, as performed in the present study. However, there are multiple methods of assessment, and we need to be aware of the differences between them. In the MSI test, five microsatellite regions (BAT-25, BAT-26, MONO-27, NR-21, and NR-24) in DNA obtained from tumor and normal tissue from the same patient are amplified using PCR [15]. Tumors are classified as having high microsatellite instability (MSI-H) if two of the five microsatellite markers present a length difference between the tumor and normal samples, low microsatellite instability (MSI-L) if only one microsatellite marker presents a length difference, and microsatellite stability (MSS) if no length difference is observed. In addition, there are NGS methods that specifically target only microsatellite regions and that evaluate MMR function as part of comprehensive cancer genome profiling approaches. When targeting microsatellite regions only, the length of a total of 18 microsatellite marker regions is measured through NGS, and MSI-H is diagnosed when 33% or more of the markers present instability [18]. Regarding MMR status, it has been reported [48] that there is a concordance rate of 90% or higher between IHC staining and MSI testing in colorectal cancer; however, another report [49] has suggested lower concordance rates in other types of cancer. In the evaluation of immunohistochemistry staining and MSI testing for endometrial cancer, the overall concordance rate was 93.3% and, in cases that were discordant, the reason was promotor hypermethylation of MLH1 [50]. Moreover, in endometrial cancer, although specific discrepancies are observed in the dMMR sub-group, IHC results are considered a better predictive factor for MMR status than determination using PCR [51]. Although IHC was performed for assessment in the current study, it is important to recognize that MSI testing has limitations that should be understood. When the DNA extraction quantity is low or the DNA quality is poor, there is a 14% probability that the test cannot provide an accurate evaluation. Furthermore, if the purity of the tumor cells in the sample is less than 30%, the results are likely to be false negative [52].

While most previous investigations in medical imaging classification have used CNNs, a combined analysis of the PORTEC randomized trial and a clinical cohort conducted by Fremond et al. [30] used attention-based models for class classification. Traditionally, CNNs have been widely used for image classification tasks; however, the introduction of the attention mechanism [53] has allowed for more accurate execution. CNNs capture relationships between adjacent pixels in images and recognize the content being displayed through structures called convolutional layers in the architecture. However, a disadvantage of CNNs is that they are influenced by elements other than the intended target, such as background objects. On the other hand, the attention mechanism, originally developed primarily for natural language processing, has also been proven to be useful in the field of image recognition. In the task of image recognition, a technology derived from Transformers [54] incorporating attention, known as a Vision Transformer, has emerged [44]. ViT models are capable of visualizing how much attention is paid to which areas within an image. Unlike CNNs, pure ViTs do not include convolutional structures and are composed solely of the attention mechanism, although they can also be used in conjunction with CNNs. Through identifying areas of interest within images using the attention mechanism, we believe that the accuracy of recognition can be improved, thus addressing the disadvantage of CNNs when combined with attention mechanisms. A further structural difference is that CNNs rely on fixed local receptive fields in the early layers, while ViTs use self-attention to aggregate global information in the early layers [55]. We compared the performance of a ViT and an original constructed API-Net-based model incorporating a CNN structure internally, as a model utilizing an attention mechanism. In this study, for the models utilizing the attention mechanism, the accuracy of the ViT on the test data set was lower compared to that of the other networks (Table 2). The reason for the lower accuracy of the ViT could be attributed to the fact that, unlike other networks, a ViT does not use convolution in its internal structure. This suggests that incorporating a CNN architecture could be beneficial for pathological image diagnosis of endometrial cancer. Additionally, in the comparison of CNNs, ResNet showed higher accuracy. Considering that ResNet is used within the API-Net-based model as the classification backbone, ResNet can be considered highly useful in this context. The potential for further performance improvement through combining CNNs with attention mechanisms may be considered for the molecular classification of cancers.

Additionally, another example of biomarkers receiving attention in the field of endometrial cancer biology is non-coding RNA (ncRNA) [56]. Endometrial cancer is significantly associated with changes in gene function mediated by ncRNA, potentially controlling cell mobility and invasion, which are important for metastasis formation, angiogenesis, resistance to chemotherapeutic agents, and the transcriptional regulation of genes. Furthermore, in the field of reproduction, ncRNAs are also known to regulate the biosynthesis and secretion of physiological sex steroids, playing significant roles in biological processes and serving as promising biomarkers for the diagnosis of reproductive disorders [57]. As shown above, machine learning has been proven to be useful for molecular biomarker estimation in several ways. The ncRNA expression profile is also a favorable candidate for features in training AI models, which are used to support endometrial cancer treatment.

H&E-stained slides are the most widely used method in the clinical context for pathologists to confirm the histological type of endometrial cancer. In this study, we confirmed that the MMR status of endometrial cancer could be predicted from H&E-stained slides using deep learning. To the best of our knowledge, there are only a few studies [27,28,30,58] worldwide that have tested this concept in endometrial cancer. ResNet, which performed particularly well in the present study, has also been used in several previous studies [27,30,31,34,58] in the field of medical imaging, although the number of layers and target organs differed. For example, in colorectal cancer, the use of artificial intelligence (AI) in the colorectal cancer diagnostic algorithm is expected to reduce testing costs and avoid treatment-related expenses [59]. A strategy using high-sensitivity AI followed by a high-specificity panel is expected to achieve the most significant cost reduction (about USD 400 million, or 12.9%) compared to that of a strategy using NGS alone [59]. Meanwhile, a strategy using only high-specificity AI may achieve the highest diagnostic accuracy (97%) and the shortest time to the initiation of treatment [59]. This report was based on cost assumptions for colorectal cancer from 2017 to 2020 in the United States. Although it is necessary to assess how much of a cost reduction can be achieved in other contexts, the use of a similar approach for endometrial cancer may have the potential to save time and costs. Additionally, the use of AI-based approaches to assist the decision-making of oncologists in treating cancer has the potential to allow optimal treatment to be provided to cancer patients sooner [60]. In this study, we constructed classification models using surgical specimens, but it is known that there is substantial agreement regarding MMR status between endometrial curettage and hysterectomy samples. Berg et al. [61] mentioned the high concordance rate that can be used not only as a value for MMR status classification in endometrial cancer but also as an independent prognostic marker before resection. Therefore, if the AI classification model could be constructed for curettage specimens, it would enable the consideration of treatment strategies at an earlier phase. However, curettage samples often exhibit significant deformations, necessitating further consideration in the construction of AI classification models.

However, there are challenges that we face in the adoption of AI for cancer treatment [62]. We need to be aware that discrepancies in international regulations create risk regarding the trustworthiness of each medical machine learning algorithm. In the actual clinical use of machine learning, interoperability and integration with existing electronic health records and image storage systems are significant barriers to adoption by hospital systems. In addition, from an ethical perspective, it is important to prioritize human dignity and protect fundamental rights such as privacy, data protection, and equality. To achieve this, it is necessary to explain the operation of AI models and systems to improve people’s understanding of them [63]. As the usefulness of AI is demonstrated in all kinds of clinical situations, we will need to continue to search for ways to solve problems surrounding medical AI for cancer treatment.

## 5. Conclusions

Molecular classification has been playing an increasingly vital role in treatment strategies for endometrial cancer. Therefore, it is crucial to incorporate additional user-friendly screening tools that can identify patients requiring further laboratory testing, thereby saving both time and costs. Our study demonstrated the potential of developing AI-based solutions, which are capable of easily and rapidly estimating molecular classifications from H&E-stained slides in clinical practice.

## Figures and Tables

**Figure 1 cancers-16-01810-f001:**
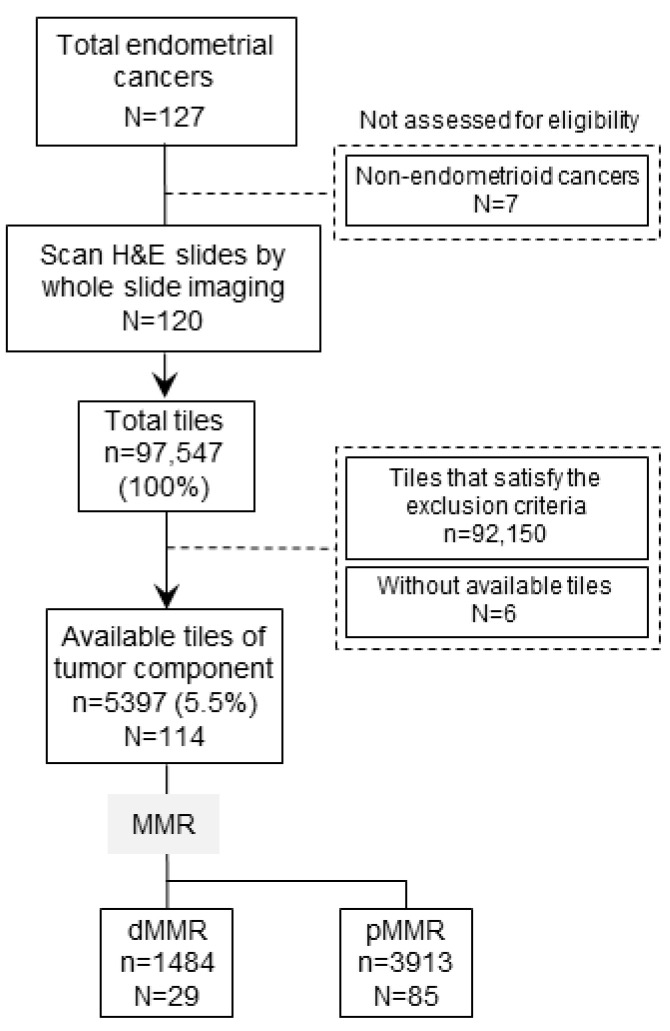
Study flow diagram. In total, 114 eligible patients were investigated in this study, and we used 5397 tiles as the study data set. N = number of patients, n = number of tiles, dMMR = deficient MMR, pMMR = proficient MMR, WSI = whole-slide imaging.

**Figure 2 cancers-16-01810-f002:**
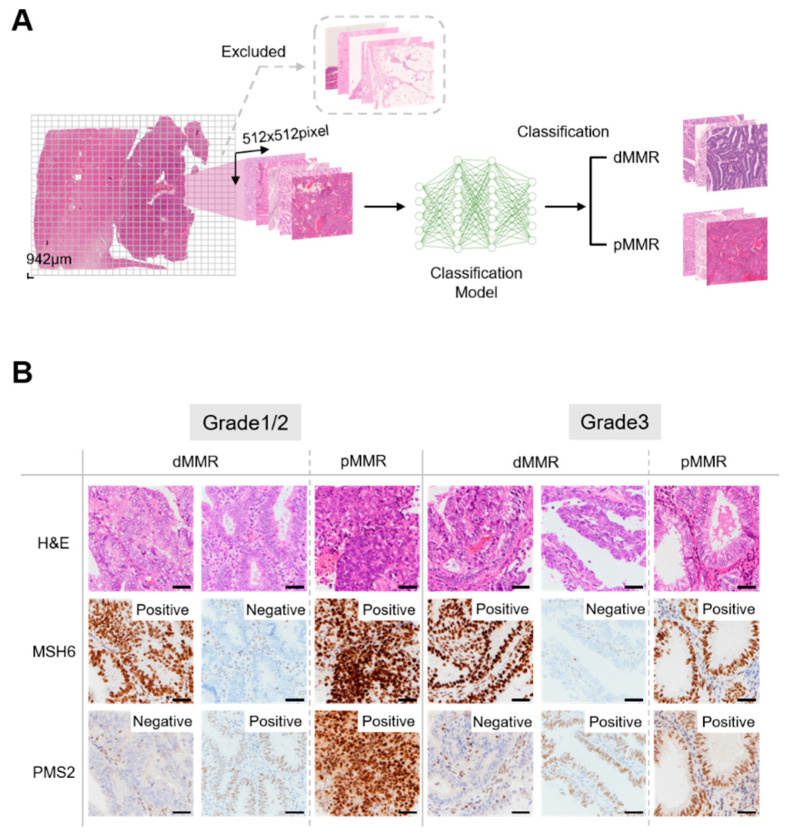
Overview of this study and evaluation of immunohistochemistry findings for MSH6 and PMS2 in endometrial cancer. (**A**) Overview of data preparation and model construction. Whole-slide images were cut into non-overlapping square tiles of 512 pixels at 5× magnification. Tiles that met the exclusion criteria were excluded (Appendix A), and only eligible tiles (Appendix A) were used as the data set. For each tile, the classification model was used to perform binary classification of mismatch repair (MMR) status. (**B**) Evaluation of immunohistochemistry findings for MSH6 and PMS2. Cases with a loss of expression in either PMS2 or MSH6 were classified as deficient mismatch repair (dMMR), while those without such loss were classified as proficient mismatch repair (pMMR). Bar = 100 μm.

**Figure 3 cancers-16-01810-f003:**
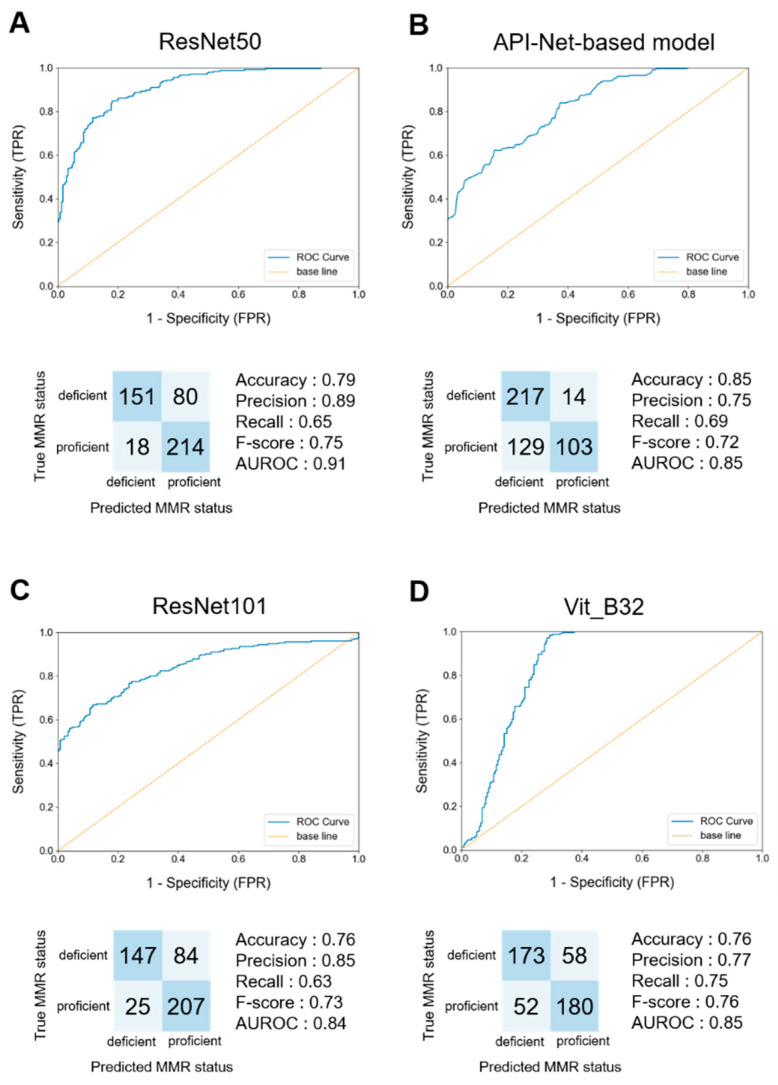
Model performance on test data set at the per-tile level. (**A**) Receiver operating characteristic (ROC) curves and confusion matrix using ResNet50. ResNet50 had the highest accuracy among the CNN models. (**B**) ROC curves and confusion matrix using API-Net-based model. The API-Net-based model had the highest accuracy among the models using the attention mechanism. (**C**) ROC curves and confusion matrix using ResNet101. ResNet101 had the second-highest accuracy among the CNN models. (**D**) ROC curves and confusion matrix using Vit_B32. Vit_B32 had the second-highest accuracy among the models using the attention mechanism. TPR = true positive rate, FPR = false positive rate.

**Table 1 cancers-16-01810-t001:** Results for metrics concerning different pre-processing of data sets and various hyperparameter configurations. We examined the ratios of the number of tiles between data sets, data augmentation, and hyperparameters.

			Accuracy	Precision	Recall	F-Score	AUROC
Pre-processing	Ratio	Original	0.74	0.55	0.09	0.15	0.74
		Down-sampled	0.80	0.76	0.88	0.81	0.89
	Data augmentation	None	0.80	0.76	0.88	0.81	0.89
		Rotate	0.76	0.70	0.90	0.79	0.87
		Flip	0.75	0.73	0.77	0.75	0.84
		Rotate and flip	0.77	0.77	0.76	0.77	0.86
Hyper-parameter	Batch	8	0.80	0.76	0.88	0.81	0.89
		16	0.72	0.70	0.74	0.72	0.80
		32	0.69	0.70	0.67	0.68	0.77
	Epoch	30	0.80	0.76	0.88	0.81	0.89
		60	0.78	0.75	0.84	0.79	0.88
		90	0.80	0.79	0.81	0.80	0.88
		120	0.75	0.69	0.89	0.78	0.87
	Learning rate	1 × 10^−2^	0.80	0.76	0.88	0.81	0.89
		1 × 10^−3^	0.70	0.68	0.75	0.71	0.78
		1 × 10^−4^	0.69	0.64	0.86	0.74	0.78

**Table 2 cancers-16-01810-t002:** Results of metrics concerning various classification models. We examined the differences in performance among classification models using CNNs or attention mechanisms.

		Accuracy	Precision	Recall	F-Score	AUROC
Convolutional neural network	GoogLeNet	0.74	0.72	0.79	0.75	0.83
	VGG_19_BN	0.79	0.86	0.68	0.76	0.85
	ResNet50	0.80	0.76	0.88	0.81	0.89
	ResNet101	0.81	0.78	0.88	0.82	0.89
	wideResNet101-2	0.77	0.88	0.62	0.73	0.88
	EfficientNet-B7	0.74	0.77	0.68	0.72	0.81
Attention mechanism	ViT_B16	0.57	0.59	0.43	0.50	0.62
	ViT_B32	0.67	0.61	0.89	0.73	0.76
	API-Net-based model	0.81	0.81	0.81	0.81	0.89

## Data Availability

The data presented in this study are available in this article (and Appendix A).

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
