# Peer review of "Prediction of Mismatch Repair Status in Endometrial Cancer from Histological Slide Images Using Various Deep Learning-Based Algorithms"

_cancers, 2024, doi:10.3390/cancers16101810_

Round 1

Reviewer 1 Report

Comments and Suggestions for Authors

Dear Authors

Thank you for giving me the opportunity to review your manuscript. I really appreciate the time and effort dedicated in this paper. The authors present a novel study with a new MMR screening with the use of AI. Congratulations for this great idea!

Material and Methods

1.     Why did the authors exclude non-endometrioid cases? MMRd is present in 10-16% of non-endometrioid histology. It should be explained in M&M. 

2.     Line 141. Have the authors considered the possibility of performing in preoperative sample? The concordance is excellent when it is compared to surgical specimen (Berg et al. 2023. doi: 10.1038/s41416-022-02063-3)

Reviewer 2 Report

Comments and Suggestions for Authors

This study's innovative use of deep learning algorithms to analyze H&E-stained endometrial cancer slides is highly promising. It serves as a beacon for the potential application of such technologies across other molecular profiles in gynecological cancers. As highlighted by Di Tucci et al., (Therapeutic vaccines and immune checkpoints inhibition options for gynecological cancers) and Zou Y  et al (Advances in the application of immune checkpoint inhibitors in gynecological tumors).immunotherapy is making significant strides in cancer treatment. By integrating artificial intelligence with emerging molecular patterns, we can not only enhance diagnostic precision but also pave the way for tailored therapeutic approaches. The successful prediction of MMR status using neural networks suggests that a similar methodology could revolutionize the way we approach other complex gynecological cancers, potentially leading to more effective and personalized treatment options. This is a crucial step towards integrating cutting-edge technology with clinical practice, aiming to significantly improve patient outcomes in the oncology field.

Comments on the Quality of English Language

Minor editing of English language is required

Reviewer 3 Report

Comments and Suggestions for Authors

Dear Authors,

It was my pleasure to review the article titled Prediction of Mismatch Repair Status in Endometrial Cancer

from Histological Slide Images Using Various Deep Learning-Based Algorithms, a rather meaningful and well-crafted analysis centered around 127 cases of the primary lesion of endometrial cancer and aimed at investigating the potential use of H&E-stained endometrial cancer slide images to predict the associated mismatch repair (MMR) status. 

The article is noteworthy and impressive, a well structured and competently assembled piece of research with strengths which cannot be overlooked: novelty, relevance in a highly sensitivr area of cancer research, and a potentially broad appeal to a relatively large scholarly readership; also worth mentioning is the fact that in terms of pursuing its stated objective, the article comes across as thorough and well-grounded in sound and clearly explicated methodology, as far as I was able to determine; the table and figures are well framed as well, and ultimately effective in outlining key data and findings. 

Some shortcomings which need to be dealt with: 

The article's objective should be enunciated more throroughly in terms of what the contributions of the study's findings would be within the broader picture of cancer care, prognostics and therapeutics. I therefore advice the authors to strive for more thorough contextualization and elaboration, especially as it pertains to crafting new innovative prognostic/therapeutic avenues stemming from personalized/precision medicine, ncRnas, molecular classifications. "Biomarkers", which is one of the keywords, are not discussed exhaustively, yet such highly innovative and still evolving avenues for diagnostics and prognostics have already been harnessed in diseases such as gynecological cancers, including EC, affecting fertility prospects, with remarkably successful outcomes and considerable prospects for further development. It is therefore worth weighing said innovations within the realm of major innovations based on as AI, machine-learning and data processing.

The Discussion is in fact the section which falls relatively short compared to the high degree of insightfulness reached with respect to the article's main objective. 

Even though the article is obviously not a review, greater elaboration is also advisable with regards to the personalized/precision medicine applications of AI, data analysis, clinical machine-learning from the standpoint of policy-making and evidence-based guidelines (which are incredibly relevant as the rate of innovation risks outpacing the standards we rely on when providing care in an equitable and fair fashion). When fleshing out such dynamics, broader contextualization is necessary, given the complexities at play. This would emphasize the value of this writing and make it more comprehensive.

The following sources should be drawn upon and cited:

Zhang R, Wesevich V, Chen Z, Zhang D, Kallen AN. Emerging roles for noncoding RNAs in female sex steroids and reproductive disease. Mol Cell Endocrinol. 2020 Dec 1;518:110875. 

Cavaliere AF, Perelli F, Zaami S, Piergentili R, Mattei A, Vizzielli G, Scambia G, Straface G, Restaino S, Signore F. Towards Personalized Medicine: Non-Coding RNAs and Endometrial Cancer. Healthcare (Basel). 2021 Jul 30;9(8):965. doi: 10.3390/healthcare9080965.

Swanson K, Wu E, Zhang A, Alizadeh AA, Zou J. From patterns to patients: Advances in clinical machine learning for cancer diagnosis, prognosis, and treatment. Cell. 2023 Apr 13;186(8):1772-1791. doi: 10.1016/j.cell.2023.01.035.

Medenica S, Zivanovic D, Batkoska L, Marinelli S, Basile G, Perino A, Cucinella G, Gullo G, Zaami S. The Future Is Coming: Artificial Intelligence in the Treatment of Infertility Could Improve Assisted Reproduction Outcomes-The Value of Regulatory Frameworks. Diagnostics (Basel). 2022 Nov 28;12(12):2979. doi: 10.3390/diagnostics12122979.

All in all, the manuscript is well-conceived, well-written and with some adjustments and a higher degree of contextualization, can provide a considerable contribution to a highly meaningful and compelling area of cancer research vis-a-vis innovations which will broaden our horizons dramatically. 

Best regards.

Comments on the Quality of English Language

Manuscript well-written overall. Further proofreading advisable.
